# OBJECTNET CAPTIONS: MODELS ARE NOT SUPERHUMAN CAPTIONERS

## ABSTRACT

Even on out-of-domain image captioning datasets such as nocaps, models often outperform humans according to captioning metrics like CIDEr. Yet, in real world conditions, model captions are often wrong. We demonstrate that this performance deficit exists by introducing a new dataset and a new captioning metric. We introduce a new dataset, called ObjectNet Captions, that reduces spurious correlations which machines often exploit. We show the shortcomings of current captioning metrics with a head-to-head experiment against humans, where we find that humans rate human-generated captions as being of much higher quality than machine captions. Driven by this, we introduce HUMANr, a new, highly robust, easy to replicate, and consistent metric, computed from head-to-head comparisons, which can be crowdsourced at low cost. We also develop tooling to automatically compute HUMANr. HUMANr is an absolute performance metric: driving it to 0 means that humans can no longer distinguish machine captions from human captions. No current metric provides such a fixed target to aim for along with knowledge of when captioning is solved in this sense. Moreover, HUMANr can reveal that humans still outperform machines, which no current metric is able to demonstrate. Existing metrics both overstate the performance of machine models and, at the same time, they inherently limit it. While most current metrics are saturated, HUMANr provides significant opportunities for further captioning research, thereby opening the door to new advances. ObjectNet Captions and HUMANr are made available to the research community.

## 1 INTRODUCTION

Machines perform remarkably well on current image captioning datasets. On nocaps out-of-domain, they significantly outperform humans (Agrawal et al., 2019; Wang et al., 2022), despite the fact that the dataset was constructed to challenge systems with novel objects. On Conceptual Captions, they are roughly on par with humans (Mokady et al., 2021; Sharma et al., 2018). Yet, anecdotally, real-world image captioning performance appears to significantly underperform. Systems routinely misidentify objects and their properties, and have nowhere near the reliability of humans. Here, we demonstrate that this gap exists by introducing a new dataset, ObjectNet Captions, and pairing it with a new evaluation methodology that overcomes a critical shortcoming in how we understand captioning performance today.

There are at least three reasons why systems perform well on current datasets but suffer when challenged by real-world conditions; each of these points is addressed by ObjectNet Captions. First, current datasets are composed of images sourced from the web which have numerous biases, such as a preference for aesthetically pleasing images. This bias largely eliminates many real-world phenomena such as clutter, conspires to put objects in common locations (such as forks in kitchens), arranges those objects in pleasing orientations (cups tend to be upright), and allows for only a few camera angles. Among existing datasets, VizWiz-Caption (Gurari et al., 2020) stands out as containing much more diverse, but not systematically debiased, images.

We build our new dataset on top of ObjectNet (Barbu et al., 2019), a dataset specifically collected to remove correlations between object class and background, object orientation, and camera angle. ObjectNet images also represent a wide range of socioeconomic conditions. ObjectNet Captions

inherits the ObjectNet license: it is only a test set and can never be used for training, ensuring that results will be more transferrable to real-world conditions.

Second, current captioning datasets tend to have short captions, with an average caption length of around 10 to 11 words (Agrawal et al., 2019). This is not because long descriptions would not be useful or because human image descriptions are naturally short. It is because annotators are asked to write about images online without incentives or guidance to produce long descriptions. Other dataset development efforts, such as Places Audio (Harwath et al., 2018), Spoken ObjectNet (Palmer et al., 2021), and Localized Narratives (Pont-Tuset et al., 2020), have observed that subjects naturally produce far richer descriptions when they speak compared to typing image captions. To that end, ObjectNet Captions consists of transcriptions of Spoken ObjectNet recordings, with an average length of 25 words.

Third, current metrics for image captioning have flaws: they both overstate and limit machine performance, holding back current research. A description system which produces flowing, accurate prose, might simply use a different style or tone than the reference captions in the dataset, resulting in poor scores. No current metric can demonstrate that machines have matched human performance (as current metrics don't have a well-defined upperbound or human setpoint) and no current metrics can enable machines to achieve parity with humans.

Our new metric, HUMANr, sidesteps these issues with current metrics. It demonstrates a large gap between machines and humans (see fig. 1), enabling new research rather than uninterpretable improvements when it is unclear how the improvements translate to caption quality. It provides a human setpoint – score 0 in HUMANr means that humans cannot distinguish

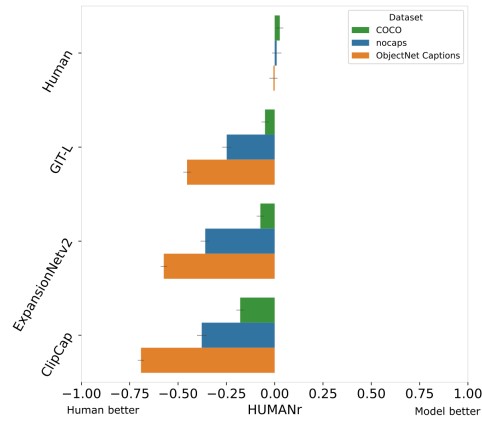

Figure 1: Human-in-the-loop evaluation demonstrating our new score, HUMANr. HUMANr measures the probability that humans prefer human or machine captions. In a head-to-head experiment, human subjects are shown an image and a pair of captions. One caption is generated by another human, the other by a model. A score of zero means that machine- and human-generated captions are indistinguishable from one another. A negative score means that humans prefer human captions, while a positive score indicates the converse. In the top row, we pit human captions against one another, and demonstrate that, as expected, they cannot be distinguished from one another. When we test machines, humans have a strong preference for human-generated captions; this is in stark contrast to what current metrics would lead one to believe. Humans prefer human captions on both nocaps and ObjectNet Captions, but the preference for human captions is almost twice as pronounced on ObjectNet Captions. The crude nature of current metrics masks the reality that humans find human-generated captions to be of far higher quality regardless of which evaluation dataset is used. Routine and automated human-in-the-loop evaluation, such as with HUMANr, can be incorporated into new research and can open the way to new methods that would otherwise languish due to the lack of headroom in current evaluation methodologies.

human and machine captions—and it allows machines to exceed human performance in a meaningful way—positive HUMANr means that humans systematically prefer machine output. While human judgments are often used in image captioning to motivate new *metrics*, what we propose is a standardized way to use human judgments to evaluate new *models*. Such a protocol has not achieved widespread adoption in the captioning community, but other areas such as speech generation often use human studies as the final arbiters of performance (Nayem & Williamson, 2021). HUMANr is simple to compute; we provide tooling to automatically run it on Mechanical Turk. It is cheap—on the order of $100—a cost that must only be paid once per submission. It is robust and replicable: with only a few hundred examples from a dataset, HUMANr stabilizes and provides a reliable performance metric.

Our contributions are: 1. a new test set, ObjectNet Captions, which consists of 100k spoken and transcribed image captions describing 20k ObjectNet images, 2. a new evaluation metric, HUMANr, which reveals a large gap between humans and machines, 3. tooling to automatically compute HUMANr, 4. a demonstration that ObjectNet Captions provides a significant challenge above that of current datasets, and 5. an analysis of the failure modes of current systems.

## 2 RELATED WORK

**Image Captioning Datasets**   Many image captioning datasets have been published in recent years (Chen et al., 2015b; Agrawal et al., 2019; Gurari et al., 2020; Yoshikawa et al., 2017; Sidorov et al., 2020; Chen et al., 2015a; Sharma et al., 2018). Much analysis to date relies on performance on COCO Captions (Chen et al., 2015b) which consists of simple captions for images from 80 object classes. Newer datasets have sought to address limitations of COCO Captions by increasing scale like Conceptual Captions (Sharma et al., 2018), including out-of-domain objects like nocaps (Agrawal et al., 2019), and in the case of TextCaps (Sidorov et al., 2020), by challenging models to retrieve particular textual information from the image. Our dataset, ObjectNet Captions, poses additional challenges to image captioning systems in both the vision and language domains. It contains out-of-distribution ObjectNet images which decorrelate objects from their typical backgrounds, orientations, and image viewpoints, each of which is paired with captions which are transcribed from spoken descriptions. In addition to the added image difficulty, these captions are significantly longer and more linguistically diverse than in previous datasets due to our spoken caption collection method. Though we continue to refer to "captions" in our dataset, the collected image-accompanying texts more closely align with the definition of "descriptions" as presented by Kreiss et al. (2021) which find that the distinction between "captions" and "descriptions" is meaningful for task definition.

**Spoken Captions**   Research in spoken language processing has led to a number of datasets consisting of images with audio captions and occasionally with corresponding text captions (Havard et al., 2017; Palmer et al., 2021; Harwath et al., 2018; Pont-Tuset et al., 2020; Oncescu et al., 2021; Monfort et al., 2021; Hsu et al., 2021; Harwath & Glass, 2015). The first such dataset to be collected on a large scale was Places Audio (Harwath et al., 2018) with 400k spoken captions. Other captioning datasets, such as SpokenCOCO (Hsu et al., 2021) and Flickr Audio (Harwath & Glass, 2015), contain both text and audio captions; however, the spoken captions were collected with annotators reading the preexisting text captions which lack the advantages of spontaneous speech. Audio captions presented in Localized Narratives (Pont-Tuset et al., 2020) were indeed collected spontaneously and demonstrate significant length and levels of detail exceeding even that of our captions. However, the task posed is to fully describe everything in the image and the typical image captioning task is to describe the most salient features in an image. Additionally, the images comprising Localized Narratives are sourced from traditional web-scraped image datasets that have been shown to contain strong biases (Torralba & Efros, 2011) and offer little challenge to state-of-the-art vision models. Spoken ObjectNet (Palmer et al., 2021) used the same data collection paradigm as Places Audio, but collected spoken captions for the 50k bias-controlled images from ObjectNet (Barbu et al., 2019).

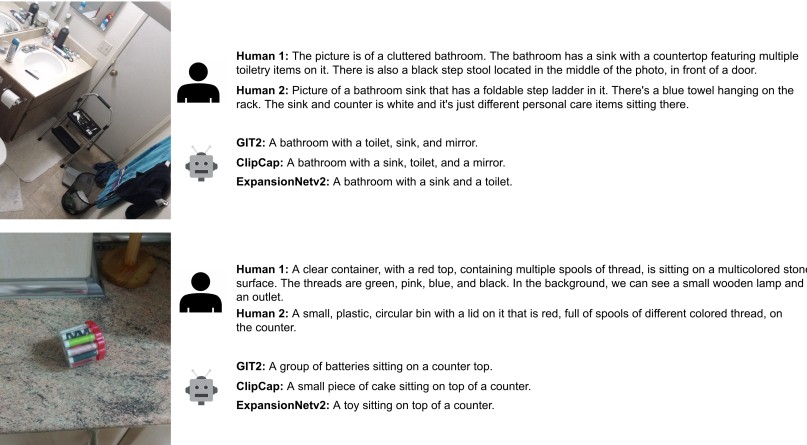

Figure 2: Example images and human-generated captions from ObjectNet Captions along with machine descriptions of those images. Human-generated captions are far richer and, as we show later, more accurate. ObjectNet Captions represents a step up in difficulty in terms of images, of text, and by having a much more rigorous automated evaluation metric that uses humans in the loop.

**Image Captioning Metrics**   Designing metrics to measure the task performance of image captioning models has been a hard problem since work on the task began (Rashtchian et al., 2010; Hodosh et al.,

2013; Kilickaya et al., 2016). Progress has been incremental and the field has largely failed to land on a convincing candidate to rally behind. The first captioning metrics were borrowed from machine translation (Papineni et al., 2002; Lin, 2004; Denkowski & Lavie, 2014), but eventually new metrics were developed specifically for image captioning (Vedantam et al., 2015; Anderson et al., 2016). Early metrics are built on rule-based text comparisons relying on increasingly sophisticated textual precision-recall tradeoffs. Failure modes for each of these metrics are well-known and well-explored (Kilickaya et al., 2016; Kasai et al., 2021).

A recognition of similarity between the tasks of caption generation and caption evaluation has led to metrics which leverage successes in language modeling and computer vision (Kusner et al., 2015; Zhang et al., 2019) including CLIP-Score (Hessel et al., 2021) which notably uses image features in its evaluation and can be used in a reference-free setting. However, this pipeline is circular. In these paradigms, we evaluate state-of-the-art models by their similarity with state-of-the-art models which confounds evaluation results and threatens to set an artificial ceiling on performance.

| Dataset | Tokens | Unique bigrams | Unique 3-grams | Unique 4-grams |
|---|---|---|---|---|
| nocaps | 9,515 | 62,799 | 121,160 | 150,132 |
| COCO | 7,794 | 49,139 | 96,711 | 124,164 |
| Ours | **13,206** | **100,081** | **226,543** | **335,113** |

Table 1: Not only are captions in ObjectNet Captions longer, but they are also more diverse. To control for the relative sizes of the datasets we randomly sample 4,500 images and 5 captions per image for each. ObjectNet Captions has nearly twice as many tokens, i.e., unique words, as COCO, and 40% more than nocaps.

Human evaluation has long been used in image captioning for evaluating models and motivating metrics (Rashtchian et al., 2010; Hodosh et al., 2013; Young et al., 2014; Aditya et al., 2015; Mitchell et al., 2012; Li et al., 2011; Vedantam et al., 2015; Anderson et al., 2016; Bernardi et al., 2016) as well as in NLP more broadly (Schuff et al., 2023; Nakache et al., 2005; Bhatt et al., 2021; Resnik & Lin, 2010). With ThumB, Kasai et al. (2021) similarly propose a human metric, although one that is not head to head—an approach that has been shown to be much more reliable (Karpinska et al., 2021). Moreover, HUMANr is much simpler to compute. Human-in-the-loop evaluation is also popular in related tasks like text-to-image synthesis (Otani et al., 2023).

## 3 DATA COLLECTION

**Images** The images for ObjectNet Captions were selected from the ObjectNet (Barbu et al., 2019) dataset. Specifically, 17,674 out of 50,273 images were chosen for ObjectNet Captions with images largely balanced across the 313 ObjectNet classes. The images were chosen to maximize the dataset's value for the task of image captioning by selecting images with longer Spoken ObjectNet (Palmer et al., 2021) captions. We reasoned that images with longer captions were more likely to be detailed or contain interesting visual features that more fully challenge captioning models.

**Spoken captions** ObjectNet Captions builds on Spoken ObjectNet 20k (SON-20k) (Palmer, 2021; Palmer et al., 2021), by collecting 5 captions per image rather than one. We followed the methodology of Spoken ObjectNet (Palmer et al., 2021) to collect spoken captions including all validation and worker qualification procedures. All told, 2,372 Mechanical Turk workers contributed spoken descriptions to ObjectNet Captions and were paid $0.25 for each task containing 4 images with an hourly wage of approximately $15 per hour.

**Transcriptions** After collecting the spoken descriptions, another Mechanical Turk task was used for transcribing the captions (Palmer, 2021). Workers were given an audio description and shown the corresponding automatic transcription in an editable textbox. Workers were instructed to listen to the recording and correct the transcription as needed. They were also instructed to add proper punctuation and capitalization as well as could be inferred. The workers could play the recording as many times as they liked and were not allowed to submit the transcription task without editing the caption. As the ASR transcription did not include any capitalization or punctuation, every caption needed at least some correction even if every word was correctly recognized. Each transcription HIT contained 4 images for which the workers were compensated $0.25 with an estimated hourly wage of approximately $15 per hour. No worker information is released with the dataset.

**Dataset Analysis** Since ObjectNet Captions is derived from spoken rather than written language, we expect that its statistics will be quite different compared to other datasets. The average caption

| Dataset | Model | B-1 | B-4 | R | METEOR | CIDEr | SPICE | BERT Score | CLIP-S | RefCLIP-S | **HUMANr** |
|---|---|---|---|---|---|---|---|---|---|---|---|
| COCO | GIT$_L$ | 80.8 ± 0.4 | **41.8 ± 0.6** | **60.3 ± 0.4** | **30.4 ± 0.3** | **136.4 ± 2.0** | 23.5 ± 0.3 | 71.9 ± 0.1 | 77.3 ± 0.2 | **82.9 ± 0.1** | -0.05 ± 0.02 |
| | ClipCap | 74.2 ± 0.4 | 32.2 ± 0.6 | 55.0 ± 0.4 | 27.1 ± 0.3 | 108.5 ± 1.8 | 20.1 ± 0.2 | 68.9 ± 0.1 | **78.3 ± 0.2** | 82.6 ± 0.1 | -0.18 ± 0.02 |
| | ExpNet | **82.7 ± 0.4** | **41.0 ± 0.6** | **60.3 ± 0.4** | 30.2 ± 0.3 | **139.6 ± 1.9** | **24.4 ± 0.2** | **73.7 ± 0.1** | 76.9 ± 0.2 | 82.7 ± 0.1 | -0.07 ± 0.02 |
| | Human | 63.1 ± 0.4 | 19.4 ± 0.5 | 46.5 ± 0.4 | 24.1 ± 0.2 | 87.8 ± 1.5 | 20.8 ± 0.3 | 58.0 ± 0.1 | **78.2 ± 0.2** | 82.2 ± 0.1 | **0.03 ± 0.02** |
| nocaps | GIT$_L$ | 74.8 ± 0.6 | **37.5 ± 0.7** | 54.2 ± 0.5 | 25.5 ± 0.3 | **94.7 ± 1.7** | 12.3 ± 0.2 | 60.6 ± 0.5 | 77.1 ± 0.2 | 82.0 ± 0.2 | -0.25 ± 0.02 |
| | ClipCap | 75.1 ± 0.4 | 29.9 ± 0.6 | 52.0 ± 0.3 | 23.8 ± 0.2 | 69.0 ± 1.5 | 10.7 ± 0.2 | 60.1 ± 0.3 | 73.1 ± 0.2 | 77.8 ± 0.2 | -0.37 ± 0.02 |
| | ExpNet | **80.3 ± 0.4** | 36.6 ± 0.6 | **55.8 ± 0.4** | 25.6 ± 0.2 | 82.2 ± 1.5 | 12.1 ± 0.2 | **62.6 ± 0.3** | 70.0 ± 0.2 | 76.5 ± 0.2 | -0.35 ± 0.02 |
| | Human | 74.8 ± 0.4 | 28.3 ± 0.5 | 52.1 ± 0.4 | **27.6 ± 0.2** | 86.4 ± 1.5 | **15.2 ± 0.2** | 58.6 ± 0.3 | **78.0 ± 0.2** | **82.6 ± 0.1** | **0.01 ± 0.02** |
| **ObjectNet Captions** | GIT$_L$ | 43.8 ± 0.3 | 16.2 ± 0.2 | 36.4 ± 0.2 | 13.3 ± 0.1 | 20.9 ± 0.4 | 8.4 ± 0.1 | 42.1 ± 0.2 | 75.8 ± 0.1 | 77.2 ± 0.1 | -0.46 ± 0.02 |
| | ClipCap | 50.0 ± 0.3 | 15.4 ± 0.2 | 35.3 ± 0.1 | 12.4 ± 0.1 | 10.2 ± 0.3 | 6.2 ± 0.1 | 39.7 ± 0.1 | 74.2 ± 0.1 | 73.7 ± 0.1 | -0.69 ± 0.01 |
| | ExpNet | 51.3 ± 0.3 | **17.6 ± 0.2** | **38.5 ± 0.1** | 13.9 ± 0.1 | 14.9 ± 0.3 | 8.1 ± 0.1 | **43.5 ± 0.1** | 72.0 ± 0.1 | 74.4 ± 0.1 | -0.56 ± 0.02 |
| | Human | **60.5 ± 0.2** | 16.1 ± 0.2 | 38.7 ± 0.2 | **20.4 ± 0.1** | **31.3 ± 0.5** | **16.3 ± 0.1** | 37.6 ± 0.2 | **77.0 ± 0.1** | **77.9 ± 0.1** | **0.0 ± 0.02** |

Table 2: Standard performance metrics and HUMANr (ours) computed for three top models along with human performance on those metrics on COCO, nocaps and ObjectNet Captions (ours). While humans lead on some metrics on ObjectNet Captions, they fall behind on others, while being significantly behind in BERTScore. Human results were produced by holding out one random caption per image. These results indicate that models and humans are comparable on this task, with models trailing humans only slightly. As we show later, this stands in stark contrast to what human evaluators think about the performance of systems and their overwhelming preference for human-generated captions. Additional metrics are available in the appendix.

length of ObjectNet Captions (25 words) is over twice as long as that of nocaps (11 words), COCO (10 words), and Conceptual Captions (10 words). This provides the opportunity to capture many more intricate and useful details of an image. The distribution of caption lengths in ObjectNet Captions has a very long tail, with a significant fraction of captions having over 30 words, which is not seen in nocaps, COCO or Conceptual Captions. The vocabulary used in ObjectNet Captions is similarly expanded; see table 1. Compared to COCO, the vocabulary is nearly twice as large, and about 40% larger than nocaps.

While ObjectNet Captions has much longer sentences, the distribution of part of speech (POS) tags is fairly similar compared to that of other datasets. Notable is the fact that the frequency of verbs is similar, despite ObjectNet images consisting almost exclusively of static scenes. Compared to other datasets, pronouns are much more frequent while nouns are somewhat less frequent. This likely makes understanding and grounding the captions considerably more difficult. See the appendix for additional dataset analysis and figures.

## 4 EXPERIMENTS

### 4.1 AUTOMATIC EVALUATION

To obtain a baseline for how current image captioning methods perform on this new dataset, we evaluated the performance of a few state-of-the-art image captioning models with COCO finetuning on ObjectNet Captions.

**GIT$_L$** (Wang et al., 2022) is a model consisting of a single image encoder and a text decoder which achieves state-of-the-art performance on many captioning benchmarks despite strong competition from models like SimVLM (Wang et al., 2021), VinVL (Zhang et al., 2021), and GRIT (Nguyen et al., 2022). The image encoder is pretrained using a contrastive vision-language framework based on (Yuan et al., 2021) and the text encoder leverages both textual input and the image encoding to generate text in a language modeling task. The text decoder is pretrained using 800M image-text pairs and then finetuned for dataset-specific tasks if necessary. We use the pretrained and publicly available GIT$_L$ model with COCO finetuning; no larger variants have been released by the GIT authors to date.

**ClipCap**(Mokady et al., 2021) is a captioning model that uses a frozen CLIP (Radford et al., 2021) model to encode images which are mapped into a textual embedding space to be used as a prefix to GPT-2 (Radford et al., 2019) to then generate text. Since both the CLIP and GPT-2 models are frozen, training burden is very light. CLIP has also been shown to be very robust to the ObjectNet distribution shift in an image classification task. We use a pretrained ClipCap model finetuned on COCO with a ViT-B/32 (Dosovitskiy et al., 2020) vision component.

**ExpansionNetv2** (Hu et al., 2022) is a recent model that uses novel expansion layers to distribute input information. The model uses a SwinTransformer (Liu et al., 2021) pretrained on ImageNet but the captioning mechanism is trained exclusively on MS COCO. Despite very limited training data with respect to the other chosen models, ExpansionNetv2 achieves impressive performance on COCO. We chose this model to provide a baseline for models without massive scale training data.

We evaluate these models using many conventional metrics: BLEU (Papineni et al., 2002), ROUGE (Lin, 2004), CIDEr (Vedantam et al., 2015), METEOR (Denkowski & Lavie, 2014), SPICE (Anderson et al., 2016), and BERTScore (Zhang et al., 2019), and CLIPScore (Hessel et al., 2021). We also collect a human baseline by computing the inter-annotator score according to each of these metrics. The results are displayed in table 2.

Beginning with ObjectNet Captions, the results of the evaluation indicate that humans outperform machines on this dataset although on most metrics, models are not far behind. The ObjectNet Captions scores reported in table 2 are generally lower than is reported on existing datasets. The human inter-annotator scores are lower than the human scores on nocaps, for example, but—with the exception of CIDEr—the scores are within the range of what we might expect from longer, more diverse captions compared to what has been reported on other datasets. The human CIDEr score on ObjectNet Captions is significantly lower than that of nocaps (86.1 versus 31.3 on ObjectNet Captions) which is likely also a result of the length and diversity of the captions. CIDEr relies on a scaled average dot-product between TF-IDF vectors (Vedantam et al., 2015); more diverse captions may have less word overlap than is found in other datasets and since our captions are longer on average, the same number of overlapping words contributes less toward the cosine-similarity.

Perhaps more interesting than what the ObjectNet Captions results show is what they do *not* show. Specifically, the results do not show high performance from models over humans as is recorded on other datasets (Wang et al., 2022). Indeed, in table 2. we see that on COCO, models outperform humans on all metrics except CLIPScore where they tie for first. Models also lead in most metrics on nocaps and where humans win, it is only by narrow margins.

We would like to highlight the CLIPScore evaluations which demonstrate what might be a concerning byproduct of model-based evaluation. By following convention and benchmarking a handful of state-of-the-art models on our dataset with established metrics, we have found ourselves in an ironic situation in which we have evaluated CLIP's similarity with itself! Indeed, ClipCap and CLIPScore both use a CLIP (Radford et al., 2021) image encoder and there is a meaningful difference in how it ranks ClipCap against our other two models. ClipCap ranks worst among the three chosen models according to every metric—including our human-in-the-loop metric—on every dataset except when measured by CLIPScore variants where it ranks often second or even first. This suggests a bias which makes it impossible to conclude anything about ClipCap's performance using CLIPScore. If such is the case then all CLIP-based models must be categorically excluded from CLIPScore evaluation which complicates benchmarking procedures by necessitating an additional metric by which CLIP-based captioning models can be measured with respect to other algorithms. However, the bias likely goes beyond just CLIP models themselves. Models that were exposed during training to data that was used to train CLIP may be preferentially evaluated by CLIPScore depending on the extent of the exposure. In the age of vison-language pretraining on web-scale data, model-based captioning metrics may introduce a subtle and nebulous data-leakage problem which is not obviously soluble. It is, however, avoidable. Human-in-the-loop evaluation sidesteps these data-leakage problems while also providing a clearer and more reliable picture of the current state of image-captioning.

## 4.2 HUMAN-IN-THE-LOOP EVALUATION

While current metrics often rank models above humans even on out-of-domain datasets like nocaps and ObjectNet Captions (see table 2), our new metric, HUMANr, does not. With ObjectNet Captions and HUMANr together, we reveal the existence and extent of the performance gap. HUMANr is a head-to-head challenge between models and machines. Participants are not told the source of any caption, only that they must select the caption which suits the image best.

We published tasks on Amazon Mechanical Turk in which a worker is shown an image from ObjectNet and two captions which describe the image. The worker is asked to indicate which description best matches the image by selecting a rating between 1 and 9. They are told that a score of 1 means that only the caption on the left can describe the image, a score of 9 means that only the caption on the right can describe the image, and a score of 5 means both captions match the image equally well. See fig. 3 for an illustration of the task setup. Images were shown up to 4 times total to different workers: Once with a randomly selected pair of its human annotated captions, three additional times with a random human annotation and a caption from each of our three models. The human-human comparisons provide a baseline variance that we use to determine whether the model-human comparisons present

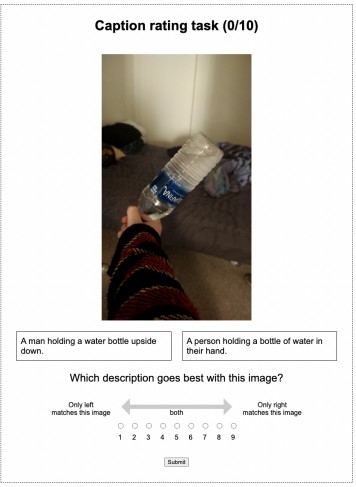
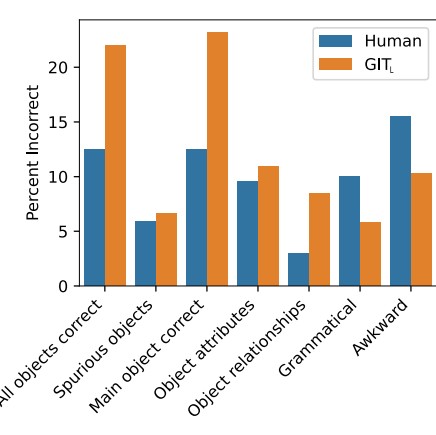

Figure 3: (left) Subjects on Mechanical Turk see an image and two captions. They choose how well the captions fit the image. Our tool automates this process enabling low-cost human-in-the-loop evaluations. (right) $GIT_L$ makes frequent visual errors like misclassifying or hallucinating objects compared to humans. Human captions are more likely to be ungrammatical or generally awkward, likely an artifact from crowdsourcing.

statistically significant deviations from human captioning performance. The order in which captions appear left-to-right on the participant's screen is randomized.

Workers complete tasks in groups of 10 with 10 distinct images. Each group contains one attention check, where one of the two captions is selected from a random image. If a worker fails the attention check by preferring that random caption, their responses are eliminated. Very few workers failed the attention checks so this did not affect our results. A total of 162 workers participated in the task. Workers were paid $0.25 per 10 comparisons receiving wages of over $15 per hour.

We ran this experiment three times: once with images and reference captions from COCO, once with nocaps, and once with ObjectNet Captions. For COCO, 5,000 images were used, for nocaps, all the images in the validation sets were used, whereas for ObjectNet Captions, a random set of about 3,500 images were selected for use.

The question posed to workers was chosen carefully to minimize the bias it injected into the task. We were careful not to imply that participants should judge the captions according to any notion of quality other than their functional role. That is, we did not ask "Which caption is better?" or "Which caption is a better description of this image?" because both of these questions are problematic: the first may encourage judgments of intrinsic caption quality that do not relate to the image at all (e.g. grammaticality), and the second is likely to bias workers toward longer captions since a reasonable interpretation of a "better description" is a more descriptive description. Instead, we asked workers simply to judge which caption "goes best" with the image or "matches" the image best and allowed them the ability to indicate whether they matched equally well.

In fig. 1, we show the average HUMANr score on each of the comparison types (Human-Human, Human-$GIT_L$, Human-ClipCap, Human-ExpansionNetv2) for each dataset. HUMANr scores are aligned such that a score of -1 indicates preference for human captions, zero indicates no preference, and a score of 1 indicates that machines are far better than humans. Our results show that, as expected, human-human results are near chance, but models underperform systematically although they have closed the human-machine gap almost entirely on COCO Captions. They do not produce human-level captions on nocaps and they struggle even more on ObjectNet Captions.

The HUMANr scores sharply diverge from the other metrics in table 2. Most other metrics indicate that models outperform humans on nocaps; whereas we see that humans consider the human captions to be better. Some metrics, like CLIPScore, slightly prefer humans but the gap between machines and humans is negligible. This leaves little room for improvement, and clearly misleads by massively understating the gap between humans and machines. Since HUMANr has a setpoint where machines equal humans, and an upper bound where they significantly exceed human performance—unlike other metrics—it provides a measure of our progress solving current datasets. With this, we can state that COCO is essentially solved which is not clear from using current metrics.

HUMANr is reproducible and robust with respect to both images and workers. The standard deviation of HUMANr computed on random half-splits of workers is very small, see fig. 4(left). As a function of the number of images evaluated, fig. 4(right), even using only 500 images leads to a HUMANr with a standard deviation of less than 0.02. HUMANr is reliable, simple, and cheap to compute.

### 4.3 EXPLAINING THE HUMAN-MODEL GAP

To understand the disparity between human caption preferences and scores from automated metrics, we selected 500 images from ObjectNet Captions. For each image, we randomly selected one $GIT_L$ caption and one human caption. We manually inspected this pair while answering seven questions: three questions about visual object classification, two questions about object properties and relationships, and 2 questions about the language used in the caption. The seven questions were: 1) Does the caption misclassify an object present in the image? 2) Does

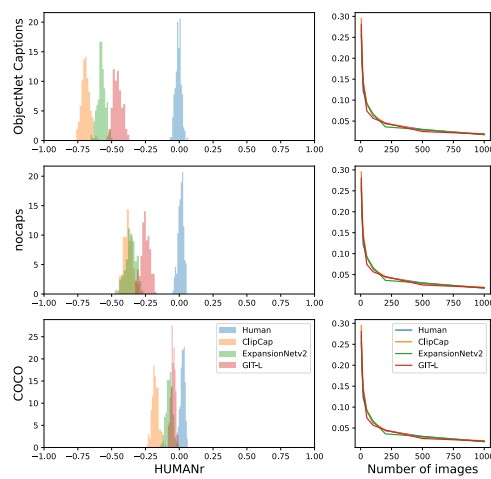

Figure 4: HUMANr reproducibility results. On the left, the distribution of HUMANr computed on randomly selected halves of the worker pool. On the right, the standard deviation of HUMANr as a function of number of images. HUMANr is robust to changes in the worker pool and quickly becomes stable with only a few hundred images.

the caption describe an object that is not present in the image? 3) Is the primary object in the image correctly classified? 4) Are all of the properties assigned to the objects correct? 5) Are all of the relationships between objects correct? 6) Is the caption grammatically correct? 7) Is the language in the caption awkward?

We find that $GIT_L$ makes significantly more visual misclassification errors than humans, almost twice as many. It also makes significantly more errors about the relationship between objects; see fig. 3. To understand the percentage of HUMANr that these questions explain, we eliminated all images where $GIT_L$ failed any of the seven checks above—human failures were not considered. This left 215 caption pairs. On the full 500 images, $GIL_L$ has a HUMANr score of -0.43, while on the restricted set it scores -0.34. This implies that these seven questions account for around 21% of the HUMANr score. While it seems like visual failures are a key part of the differences between humans and machines, the root cause of most failures is still unclear.

## 5 DISCUSSION AND LIMITATIONS

With the combination of our new dataset and new metric, HUMANr, we present a starkly different picture of the state-of-the-art in image captioning. Current metrics indicate that machines either exceed human performance or that they barely underperform. Our metric reveals that machines vastly underperform humans on nocaps. We also release ObjectNet Captions which presents a tougher challenge with images that lack many spurious correlations, text which is longer and more detailed, and a novel metric. We hope that this large performance gap will support new research directions which would otherwise not be possible with metrics that had saturated. The benefits to adopting better metrics are immense: a clearer picture of where we are and where we must go in the future.

While model-based metrics like CLIPScore seem on the surface to improve on the weaknesses of existing metrics, our results show that we should be concerned about their circularity. They seem to prefer models like themselves and even when such metrics show a human-machine gap, it is small and inconsequential. This makes research difficult; there is not much to improve. We encourage research that investigates the existence and extent of these biases.

Likely, the most controversial part of our approach is our promotion of a new metric that uses human-in-the-loop evaluation. The general position of the community is that human evaluation is optimal but intractable while automatic metrics are all too convenient to rethink. We challenge this view. Our work shows that automatic metrics are perhaps more problematic than is believed—especially with

the potential biases in increasingly popular machine metrics—while also demonstrating that human evaluation is cheaper, easier, and more reproducible than is believed. While traditional methods have guided image captioning toward impressive achievements, they are becoming obsolete. As models become increasingly capable of visual understanding, our tools for evaluating them must become increasingly valid and accurate.

As for the reliability of MTurk, running HUMANr with the same input will give slightly different results, but that does not mean it is not reproducible. For most sciences, especially those which rely on measurements of human behavior—which we believe is true of AI—this is not how reproducibility is defined (Resnik & Shamoo, 2017). Extensive surveys have been performed to investigate the repeatability of experiments conducted on MTurk concluding that though there are challenges, the problems are manageable, solutions exist, and we should welcome crowdsourcing as a legitimate scientific practice (Hauser et al., 2019; Stewart et al., 2017). Many classical results in cognitive science are easily reproduced on MTurk, even between different worker populations (Stewart et al., 2017). Our own results demonstrate that this holds true of our caption comparison task (fig. 4). Running HUMANr with enough images to ensure stability and reporting scores with confidence intervals will produce reproducible results and enable much-needed improvements to caption evaluation.

To alleviate concerns about cost and tractability of human evaluation we release a tool to automate this process on Mechanical Turk [1]. This eliminates many of complexities and concerns around reproducibility by standardizing HUMANr. Computing HUMANr is also fairly cheap — around $100 for one model on one dataset. This is affordable to most research groups and provides an evaluation that cannot be matched by current metrics which can only crudely measure overlap with existing annotations and cannot identify other mistakes in generated captions.

While human-in-the-loop evaluation metrics for determining the performance of systems have not gained traction in the computer vision community, we hope to normalize this situation in the future. Human evaluation is already commonplace in speech synthesis (Hayashi et al., 2019; Nayem & Williamson, 2021) and is growing more common for image generation (Otani et al., 2023).

The nominal cost of human-in-the-loop metrics may also on its own be beneficial. Overfitting to metrics is a major problem throughout machine learning. This small cost makes it much more likely that evaluations will be used as intended. Rather than parameter sweeping to carefully fine-tune near arbitrary constants in order to beat a benchmark, HUMANr is much more likely to be used in the final stages to validate a model.

Only a small random subset of any one dataset need be evaluated to establish a HUMANr score. A few hundred images drive the variance in HUMANr very close to zero. This makes HUMANr replicable, easy to compute, and cheap. Although any official report of HUMANr should be computed using a large enough sample to ensure stability with reasonable error bounds, unofficial runs need not be costly. Spending $10 here and there to get a signal during model development could be a cost effective way to gauge progress. Such human-in-the-loop feedback cycles during training have shown promise with recent advances in language modeling (Ouyang et al., 2022).

Although we motivate our own metric for adoption, we do not argue that automatic metrics need be abandoned. We maintain that HUMANr should be reported along with each new model release, but that automatic metrics will still prove generally useful—especially in model development where metrics can efficiently provide gradients for model updates. We encourage researchers, however, to consider these metrics as a means to an end rather than an end in themselves.

While ObjectNet Captions presents a meaningful challenge to today's captioning models, it has its limitations. The dataset contains only objects and scenes which are commonly found in the home and does not cover important domains such as natural scenes, people, etc. ObjectNet Captions could support other tasks in the future such as an end-to-end task that describes images directly in audio without intermediate transcriptions. Features only present in the raw audio files such as the timing of words, the spacing between words, and the emphasis placed on each word make this dataset uniquely useful for evaluating models on an image to spoken description task compared to past text-only captioning datasets. Extending this methodology and dataset to visual question answering, a field that has had many issues with evaluation, is of great interest to us.

---

[1]This code toolkit is in the supplemental material and will be released on GitHub upon acceptance

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
