# OpenReview forum: "ObjectNet Captions: Models are not superhuman captioners"
_ICLR.cc/2024/Conference — Submitted to ICLR 2024_

### Official Review · Reviewer_DRVD · 2023-10-31

**Soundness:** 3 good
**Presentation:** 3 good
**Contribution:** 2 fair
**Rating:** 6
**Confidence:** 3

**Summary:**

This paper introduces ObjectNet Captions, a dataset created to mitigate the exploitation of spurious correlations by machine learning models in image captioning tasks. Along with this dataset, the authors present HUMANr, a novel captioning metric aimed at providing a robust and consistent measure of performance that can be easily replicated and crowdsourced. HUMANr is intended to be an absolute performance metric that provides a clear target for model improvement and the ability to recognize when human-level captioning has been achieved, addressing the overestimation of machine performance by current metrics.


# Post-rebuttal
I appreciate the efforts made by the author. Their responses partially address my concern about the comparison and the scale of dataset. Therefore, I raise my rating. I encourage the author to make their proposed dataset and metric easily to use, such as can easily download and running via `pip`, to let people use them in practical ways.

**Strengths:**

There are several strengths for this paper:

- Introduction of a new dataset that targets a key issue, specifically the reliance on spurious correlations by captioning models.
- Development of HUMANr, and it can be easily implemented and crowdsourced.
- Potential to recalibrate the understanding of machine captioning performance, as HUMANr contrasts with existing metrics by showing the superiority of human captions.
- The paper provides tools for automatic computation of HUMANr (in supplementary), facilitating its adoption by the research community.
- It examined several learning-based Captioning models and metrics.

**Weaknesses:**

I feel there are two major flaw points:

- The authors currently did not use GPT-related captioning models, such as BLIP2. According to my usage, BLIP2 outperforms the compared methods used in this paper.

-  The proposed dataset only contains 17,674 images which are quite small-scale to evaluate a captioning model comprehensively.

**Questions:**

Please address the concerns mentioned above.

Could the author please also provide random sampled image-captions pairs. The current appendix only contains a few examples which cannot be assessed comprehensively.

---

> ### Author Response · Authors · 2023-11-21
> **Thank you for your review!**
>
> Thank you for your review and appreciation of our contributions.
>
> >The authors currently did not use GPT-related captioning models, such as BLIP2. According to my usage, BLIP2 outperforms the compared methods used in this paper.
>
> In response to reviewers’ comments, we evaluated the publicly available 6.7B parameter BLIP-2 model on ObjectNet Captions, which is an LLM-based captioning model. It received a HUMANr score of -0.41 +/- 0.03 which places it ahead of GIT_L but far below human performance. This is further evidence of the arguments we articulate in our paper: ObjectNet Captions is a challenging dataset and HUMANr is invaluable in its ability to quantify model performance in comparison to humans. We will revise our paper to include this result.
>
> Although very promising, the recent LLaVA models are—according to the ICLR reviewing guidelines—considered contemporaneous with our submission (checkpoints and peer-review were released <4 months ago). Although this excuses us from engaging with the models, we will include them as compute power and time to run on mechanical turk allows. GPT4V is difficult to evaluate because OpenAI doesn't allow new signups and throttles current accounts to only 100 images per day. We are working on including both in the final submission. LLaVA so far does show promise and our preliminary results say that it does close some of the gap with humans. Having established ObjectNet Captions and HUMANr as effective tools for measuring captioning performance, we can apply these models in future work to exploring and measuring the performance gains of these recent models.
>
> As models become increasingly good, automatic metrics will fail to reliably measure just how good they are. Indeed in preliminary results, GPT4v scores very poorly across all the automatic metrics we report in our paper despite performing much better in HUMANr. We need to explicitly ground model evaluation in systematic human judgment in order to be able to quantify this progress. It may be obvious anecdotally that GPT4v is an improvement over other methods, but HUMANr allows us to quantify that improvement in a way that is aligned with our conception of what qualitative improvement is.
>
> >The proposed dataset only contains 17,674 images which are quite small-scale to evaluate a captioning model comprehensively.
>
> The dataset is rather small, but we don’t believe that is a significant limitation. Because ObjectNet Captions inherits the ObjectNet license, it cannot be used to train models so its purpose is only to evaluate which we recommend is done using our new metric HUMANr. As shown in fig 4 in our paper, the variance in HUMANr score declines sharply with the number of images used in the evaluation. Even just 500 images drives the standard deviation very close to zero. If anything, with 17,674 images, ObjectNet Captions is much larger than it needs to be to report reliable HUMANr evaluation. It should also be noted that our dataset is not small compared to other captioning test sets. For example, the nocaps test set contains only 10,600 images.
>
> >Could the author please also provide random sampled image-captions pairs. The current appendix only contains a few examples which cannot be assessed comprehensively.
>
> Our apologies, because of the constraints on the size of the supplemental upload, we are unable to add any additional examples of images. However, you can run following script to visualize image/caption pairs. It requires downloading the ObjectNet dataset and using the ObjectNet Captions JSON file in the supplemental material.
>
> ```
> import os
> import PIL
> import json
> import glob
> import random
> from matplotlib import pyplot as plt
>
> OBJECTNET_DIR = # path to objectnet images directory
> images = glob.glob(OBJECTNET_DIR)
>
>
> CAPTIONS_FILE = # path to objectnet captions json
> with open(CAPTIONS_FILE, 'r') as f:
>     captions = json.load(f)
>
> image_to_captions = {}
> for x in captions:
>     img = x['image']
>     caption = x['caption']
>     if img not in image_to_captions:
>         image_to_captions[img] = []
>     image_to_captions[img].append(caption)
>
> images = [img for img in images if os.path.join(*img_full.split('/')[-2:]) in image_to_captions]
>
> random.shuffle(images)
> img_full = images[0]
> img = os.path.join(*img_full.split('/')[-2:])
> plt.imshow(PIL.Image.open(img_full));plt.show()
> for i, cap in enumerate(image_to_captions[img]):
>     print(f'{i}: {cap}')
> ```

---

### Official Review · Reviewer_zcyp · 2023-11-02

**Soundness:** 2 fair
**Presentation:** 3 good
**Contribution:** 2 fair
**Rating:** 5
**Confidence:** 4

**Summary:**

To evaluate the captions generated by machines, this paper collected a dataset and proposed a new human study protocol. The machine-generated captions are compared with human-generated captions and humans are involved in the evaluation loop. The human study is performed on three datasets, i.e., COCO, Nocaps, and ObjectNet Captions. Three models are evaluated in this experiment, i.e., GIT, ClipCap, ExpNet.

**Strengths:**

This paper focuses on an important problem for the image captioning community, i.e., how big is the difference between machine generate captions and human-generated captions.
The conclusion that the machine-generated captions still underperform human-generated captions on unusual datasets and fail to generate long sentences is insightful for the community.

**Weaknesses:**

However, there are several unclear questions need clarification.

1. Apart from revealing how big is the difference between machine-generated captions and human-generated captions, it would be meaningful to reveal what is the difference between machine-generated captions and human-generated captions. Though the authors have revealed some differences, such as spurious objects and caption lengths, the root cause seems still unclear.

2. Some experiment details are missing. For instance, how to compute the HUMANr score?

3. Asking human participants to rate between 1-9 seems subjective. If two new image captioning models are evaluated with two different groups of people, will the results be comparable? It would be interesting to show the deviation of two different groups of people rating the same model in Figure 4.

4. The ObjectNet cannot be regarded as a contribution as the authors only select some images with longer captions.

**Questions:**

In Section 4.3, paragraph 2, what does ``we eliminated all images where GITL failed any of the seven checks above—human failures were not considered’’ mean?

---

> ### Author Response · Authors · 2023-11-21
> **Thank you for your review!**
>
> Thank you for your appreciation. We also think this is an important issue for the image captioning community.
>
>
> >However, there are several unclear questions need clarification.
>
> >Apart from revealing how big is the difference between machine-generated captions and human-generated captions, it would be meaningful to reveal what is the difference between machine-generated captions and human-generated captions. Though the authors have revealed some differences, such as spurious objects and caption lengths, the root cause seems still unclear.
>
> We agree! We are also interested in what differences exist between model and human captions such that human evaluators prefer human captions. In Section 4.3, we choose 7 dimensions to evaluate model and human captions on and do some controlled analysis. What we find is that models are more likely to misclassify objects, misattribute properties to objects, and misjudge the physical relationships between objects. However, all together, our 7 dimensions only account for 21% of the HUMANr score which underscores the difficulty of explicitly characterizing the reasons for the gap in performance. We are interested in working on this in future work, however. HUMANr provides a good way to establish human qualitative judgments that can be investigated to determine features that may explain the judgments.
>
> >Some experiment details are missing. For instance, how to compute the HUMANr score?
> Asking human participants to rate between 1-9 seems subjective. If two new image captioning models are evaluated with two different groups of people, will the results be comparable? It would be interesting to show the deviation of two different groups of people rating the same model in Figure 4.
>
> We apologize for not explicitly describing the calculation of HUMANr. As described in the text, each task is given a rating of 1-9 by a human worker where 1 means only the caption on the left matches the image, 9 means only the caption on the right matches the image, and 5 means the two captions match equally well. That is, the magnitude of the value 1-9 should be interpreted as a slider to indicate which side to give preference to (1 and 9 are equally valent, just on opposite sides). This interpretation is made very clear to the workers.
>
> Then we transform the scores such that each caption is scored as if the human caption were on the left. So if a task with a model caption on the left and human caption on the right was given a score of 7, it would be transformed to a 3. Now that all the scores have the same alignment, we average them all together and then normalize to [-1, 1] where a negative score implies the human captions were preferred on average and a positive score means the model captions were preferred on average.
>
> As figure 4 demonstrates, the difference in HUMANr score between random groups of workers is very small in our experiments, even when those groups are disjoint. Even though individual workers may have subjective preferences, when aggregated over hundreds or even thousands of images, the HUMANr score is quite stable and robust to the exact makeup of the worker pool.
>
> >The ObjectNet cannot be regarded as a contribution as the authors only select some images with longer captions.
>
> We feel that by augmenting ObjectNet and Spoken Objectnet with transcriptions, we have enhanced its usefulness and made a significant contribution to the image captioning field.
>
>
> >In Section 4.3, paragraph 2, what does ``we eliminated all images where GITL failed any of the seven checks above—human failures were not considered’’ mean?
>
> This means that we took all the captions that we investigated for which GIT_L passed all 7 of the questions in our quality assessment, but we did not filter the human captions. The result is an optimistic comparison (in favor of GIT_L) between GIT_L captions that didn’t fail the assessment on any dimension versus human captions that may have failed some dimension of the assessment. This shows that even when we consider GIT_L at its best against fallible humans, it still significantly underperforms a human level in terms of HUMANr score.

---

> ### Comment · Reviewer_zcyp · 2023-11-22
>
> Thanks to the authors for the reply. Still, my major concerns are not well-resolved.
>
> This paper investigates an interesting topic but the experiment design and analysis still need polish.
>
> I would like to keep my original rating.

---

> > ### Author Response · Authors · 2023-11-22
> >
> > Thank you for considering our responses. We would like to do what we can to resolve further concerns you have.  We feel as though we’ve thoroughly explained and motivated our experimental design and analysis in detail. We feel our dataset, metric, and analysis of existing models will be of great benefit to the research community and we would appreciate specific suggestions for improvements as we plan to continue building on this work. What specific concerns do you have remaining and in what way were our responses to your concerns unsatisfactory?

---

### Official Review · Reviewer_X1XD · 2023-11-04

**Soundness:** 3 good
**Presentation:** 3 good
**Contribution:** 2 fair
**Rating:** 5
**Confidence:** 4

**Summary:**

This paper focuses on the task of image captioning and proposes a new dataset and a new metric. There are some findings, for example, there is a large gap betten human and models on the task of image captioning. The proposed dataset if challenging compared with existing ones, which contains much more unique tokens and n-grams and should be useful for the community.

**Strengths:**

1. a new dataset is proposed. The dataset is more challenging and contains more unique tokens and n-grams.
2. a new metric is proposed.
3. analysing existing models vs. human using a wide range of metrics.

**Weaknesses:**

1. the scale of the dataset is small.
2. the auther only considers traditional image captioning models. Some LLM-based models like LLaVA should be considered and the comparison among these models should be more interesting.
3. the findings that there is a large gap betten human and models is a common sense, so I do not think it is a significant contribution. But if the author can show that the most advanced models like GPT-4v is inferior to humans and the proposed metric is able to measure the gap, it should be more interesting.

**Questions:**

Some important references related to image captioning metrics are missing.
1. Learning to evaluate image captioning. CVPR 2018.
2. Describing like humans: on diversity in image captioning, CVPR 2018.
3. On diversity in image captioning: metrics and methods, TPAMI, 2022.

---

> ### Author Response · Authors · 2023-11-21
> **Thank you for your review!**
>
> Thank you for your comments!
>
> >the scale of the dataset is small.
>
> The dataset is rather small, but we don’t believe that is a significant limitation. Because ObjectNet Captions inherits the ObjectNet license, it cannot be used to train models so its purpose is only to evaluate which we recommend is done using our new metric HUMANr. As shown in fig 4 in our paper, the variance in HUMANr score declines sharply with the number of images used in the evaluation. Even just 500 images drives the standard deviation very close to zero. It should also be noted that our dataset is not small compared to other captioning test sets. For example, the nocaps test set contains only 10,600 images.
>
>
> >the auther only considers traditional image captioning models. Some LLM-based models like LLaVA should be considered and the comparison among these models should be more interesting.
>
> Although very promising, the LLaVA models are—according to the ICLR reviewing guidelines—considered contemporaneous with our submission (checkpoints and peer-review were released <4 months ago). Although this excuses us from engaging with the models, we will include them as compute power and time to run on mechanical turk allows. GPT4V is difficult to evaluate because OpenAI doesn't allow new signups and throttles current accounts to only 100 images per day. We are working on including both in the final submission. LLaVA so far does show promise and our preliminary results say that it does close some of the gap with humans. Having established ObjectNet Captions and HUMANr as effective tools for measuring captioning performance, we can apply these models in future work to exploring and measuring the performance gains of these recent models.
>
> As models become increasingly good, automatic metrics will fail to reliably measure just how good they are. Indeed in preliminary results, GPT4v scores very poorly across all the automatic metrics we report in our paper despite performing much better in HUMANr. We need to explicitly ground model evaluation in systematic human judgment in order to be able to quantify this progress. It may be obvious anecdotally that GPT4v is an improvement over other methods, but HUMANr allows us to quantify that improvement in a way that is aligned with our conception of what qualitative improvement is.
>
>
> >the findings that there is a large gap betten human and models is a common sense, so I do not think it is a significant contribution.
>
> We know that the gap in performance between humans and models in image captioning is no secret. Every researcher knows anecdotally that models fail in ways that humans don’t. However, until now, there has been no way to measure that gap. Automatic metrics cannot quantify this gap for us. Rather, if we relied solely automatic metrics and not on anecdotal evidence, we would be led to believe that the gap does not exist. We introduce HUMANr not to prove that the gap exists—like the reviewer notes, everyone knows that the gap exists—but rather to present a method for measuring that gap. Our contribution is not discovering that gap, but rather presenting and motivating a metric which actually reflects that gap which has not been done systematically in the literature. We will revise the language of the text to reflect this.
>
> >But if the author can show that the most advanced models like GPT-4v is inferior to humans and the proposed metric is able to measure the gap, it should be more interesting.
>
> Please see above for discussion regarding GPT4v limited availability.
>
> >Some important references related to image captioning metrics are missing.
>
> Thank you for these references! We will be sure to include them in the camera ready revision.

---

### Official Review · Reviewer_V5WD · 2023-11-13

**Soundness:** 2 fair
**Presentation:** 2 fair
**Contribution:** 2 fair
**Rating:** 5
**Confidence:** 3

**Summary:**

This paper introduces "ObjectNet Captions," a challenging dataset for image captioning, and presents HUMANr, a new metric for evaluating caption quality. It highlights a significant performance gap between human and model-generated captions, emphasizing the limitations of current models in generating detailed, accurate captions. The study's findings challenge the notion that advanced models like GPT-4 surpass human capabilities in this domain.

**Strengths:**

1. Introduction of a challenging dataset and HUMANr metric.
2. In-depth comparison of existing models with human performance using various metrics.
3. The paper effectively showcases the limitations of current models in handling diverse and complex captioning scenarios.

**Weaknesses:**

1. The dataset's focus on home environments and its relatively small size (17,674 images) may limit its generalizability.
2. Not including state-of-the-art models like BLIP2 or LLM-based models in the analysis.
3. The human-centric approach, while insightful, may introduce new biases and subjectivities.
4. The cost and scalability of HUMANr in large-scale applications are not addressed.
5. The revelation of a performance gap between humans and models is not a novel insight and lacks depth without comparing the most advanced models.
6. The paper omits crucial experimental details, like the computation of HUMANr and handling discrepancies in human evaluations.

**Questions:**

1. How can the ObjectNet Captions dataset be expanded to cover a broader range of environments and scenarios?
2. What steps can be taken to include state-of-the-art models like BLIP2 in future evaluations?
3. How does HUMANr address the subjectivity and potential bias in human judgment?
4. Are there plans to adapt the dataset and HUMANr for non-English languages or diverse cultural contexts?
5. How can the scalability and cost-effectiveness of HUMANr be improved for widespread adoption?
6. Can the authors provide more details on the methodology, especially regarding the computation of HUMANr and the management of subjective human ratings?

---

> ### Author Response · Authors · 2023-11-21
> **Thank you for your review! (Part 1)**
>
> >Introduction of a challenging dataset and HUMANr metric.
> In-depth comparison of existing models with human performance using various metrics.
> The paper effectively showcases the limitations of current models in handling diverse and complex captioning scenarios.
>
> Thank you for your comments on the strengths of our paper.
>
> >The dataset's focus on home environments and its relatively small size (17,674 images) may limit its generalizability.
>
> The restriction to in-home environments is a limitation, but it still allows ObjectNet to meaningfully probe models abilities. The in-home/indoor environment is an important context where many robots or future AI systems could be deployed. It provides a wide variety of objects and significant visual complexity. Also, non-expert human annotators have the appropriate visiolinguistic experience to caption indoor in-home images which makes captioning easy. Future work could explore captioning in other environments, but to start with, the indoor environment is likely the most broadly generalizable and the most difficult.
>
> As for the size of ObjectNet Captions, we don’t believe its size is a limitation. As shown in fig 4 in our paper, the variance in HUMANr score declines sharply with the number of images used in the evaluation. Even just 500 images drives the standard deviation very close to zero. It should also be noted that our dataset is not small compared to other captioning test sets. For example, the nocaps test set contains only 10,600 images.
>
>
> >Not including state-of-the-art models like BLIP2 or LLM-based models in the analysis.
>
> In response to reviewers’ comments, we evaluated the publicly available 6.7B parameter BLIP2 model on ObjectNet Captions. It received a HUMANr score of -0.41 +/- 0.03 which places it ahead of GIT_L but far below human performance (negative is worse than humans, 0 is human level, and positive is better than humans; with a range of -1 to +1). This is further evidence of the arguments we articulate in our paper: ObjectNet Captions is a challenging dataset and HUMANr is invaluable in its ability to quantify model performance in comparison to humans. We will revise our paper to include this result.
>
> As for recent LLM-based methods, the LLaVA models are very promising but are—according to the ICLR reviewing guidelines—considered contemporaneous with our submission (checkpoints and peer-review were released <4 months ago). Although this excuses us from engaging with the models, we will include them as compute power and time to run on mechanical turk allows. GPT4V is difficult to evaluate because OpenAI doesn't allow new signups and throttles current accounts to only 100 images per day. We are working on including both in the final submission. LLaVA so far does show promise and our preliminary results say that it does close some of the gap with humans. Having established ObjectNet Captions and HUMANr as effective tools for measuring captioning performance, we can apply these models in future work to exploring and measuring the performance gains of these recent models.
>
> As models become increasingly good, automatic metrics will fail to reliably measure just how good they are. Indeed in preliminary results, GPT4v scores very poorly across all the automatic metrics we report in our paper despite performing much better in HUMANr. We need to explicitly ground model evaluation in systematic human judgment in order to be able to quantify this progress. It may be obvious anecdotally that GPT4v is an improvement over other methods, but HUMANr allows us to quantify that improvement in a way that is aligned with our conception of what qualitative improvement is.

---

> ### Author Response · Authors · 2023-11-21
> **Thank you for your review! (Part 2)**
>
> >The human-centric approach, while insightful, may introduce new biases and subjectivities.
>
> There are, of course, some trade-offs to be made when deciding to use human evaluation.
>
> Although we note that any human annotators bring their own biases to every experiment, we attempted to make instructions as clear as possible to maximize reproducibility. The fact that workers agree on HUMANr (as evidenced by the fact that the variance is small / inter-coder agreement is high) means that workers understood the instructions and answered consistently. AMT is spread around the world, its population changes radically across continents during a long experiment, we would not have this level of consistency if each group was strongly influenced by local biases. There may be AMT-specific biases that are shared by all workers, but there is little evidence for this.
>
> Even if human workers were to be biased in a particular way, there’s no clear reason why we would want to remove this bias. Parity with human capability is, after all, our goal. We want to build models that are as good as humans as judged by humans. If a model scores 0 HUMANr, that means that a large group of humans did not find its captions to be distinguishable on average from human captions. That is no small feat regardless of what bias you think might exist in the individual worker subjectivity. In future work, as models move closer to human level performance on this general captioning task our same method could be used to study vision tasks with more culturally subjective responses and the background of human evaluators could then be taken into account when aggregating results.
>
> >The cost and scalability of HUMANr in large-scale applications are not addressed.
>
> We address the cost and scalability of HUMANr in the discussion section of our manuscript. Was there a particular portion of our discussion that the reviewer found insufficient?
>
> It does, of course, cost money to use human evaluation, but we feel that the costs are outweighed by the benefits. HUMANr gives us a measurement of captioning performance that is grounded in qualitative assessment by humans. Automatic metrics are cheap and easy, but they give us no guarantees about how quantitative improvement translates to qualitative improvement. We don’t intend HUMANr to replace automatic metrics completely, however. Automatic metrics can be efficiently computed during training and have the added advantage of being differentiable. HUMANr is primarily intended to be used for final evaluation as reported in a paper when presenting a model. As we have shown, this final evaluation can be quite cost efficient as even just 500 images drives HUMANr variance close to zero. This small (but not nonexistent) cost can even be a good thing as it discourages excessive hyperparameter tuning to overfit to metrics.
>
> For large applications in high-resource labs, there may be big advantages to integrating HUMANr into training pipelines. Indeed, recent improvements using reinforcement learning with human feedback have demonstrated this kind of approach to be very fruitful. HUMANr could be computed while training large models to get a reliable performance signal. This could be very expensive, but for labs with the funds to do RLHF, it could lead to substantial improvements.
>
> Basically, HUMANr is very cheap to use in its originally intended application, but could also be scaled up if so desired.

---

> ### Author Response · Authors · 2023-11-21
> **Thank you for your review! (Part 3)**
>
> >The revelation of a performance gap between humans and models is not a novel insight and lacks depth without comparing the most advanced models.
>
> We know that the gap in performance between humans and models in image captioning is no secret. Every researcher knows anecdotally that models fail in ways that humans don’t. However, until now, there has been no way to measure that gap. Automatic metrics cannot quantify this gap for us. We introduce HUMANr not to prove that the gap exists, but rather to present a method for measuring that gap which we show is not measured by automatic metrics and is only anecdotally recognized. We will revise the language of the text to reflect this.
>
> As for analysis with the most advanced models, please refer to our response above. Models such as BLIP2 have improved in their HUMANr score, and we plan to include even more advanced models on a leaderboard website as those become available. However, automatic metrics fail to measure this performance improvement which further supports the need to systematize human-in-the-loop evaluation like we do with HUMANr.
>
> >The paper omits crucial experimental details, like the computation of HUMANr and handling discrepancies in human evaluations.
>
> As described in the text, each task is given a rating of 1-9 by a human worker where 1 means only the caption on the left matches the image, 9 means only the caption on the right matches the image, and 5 means the two captions match equally well. We randomize which caption is on the left and right so our rating scale requires a simple transformation before we can aggregate the scores such that each caption is scored as if the human caption were on the left. So if a task with a model caption on the left and human caption on the right was given a score of 7, it would be transformed to a 3. Now that all the scores have the same alignment, we average them all together and then normalize to [-1, 1] where a negative score implies the human captions were preferred on average and a positive score indicates that the model captions were preferred on average. We will include this more detailed explanation in the revision.
>
> As for discrepancies in human evaluations, each task contains an attention check, as described in our paper on page 7, in which a caption from a random image was added to the task. If the worker fails the attention check by selecting the random caption, all of their responses to any task are ignored when computing HUMANr. That way, we ensure that all our results come from attentive workers.
> We also show in figure 4 that the HUMANr score variance between disjoint sets of the worker pool is very low. When aggregated over hundreds of images, the subjectivity of individual workers is largely averaged out resulting in a convergent HUMANr score with low variance.
>
> >How can the ObjectNet Captions dataset be expanded to cover a broader range of environments and scenarios?
>
> The spoken caption collection paradigm used to generate captions for ObjectNet Captions is applicable to any dataset. Our work demonstrates the value of using both more complex images and transcribed spoken language to collect more thorough captions. We would love for others to expand on our work and we have made this as easy as possible by releasing a toolkit for collecting HUMANr judgments and basing our captioning method on the publicly available spoken captions toolkit.
>
> >What steps can be taken to include state-of-the-art models like BLIP2 in future evaluations?
>
> See above for discussion regarding more recent model evaluations.
>
> >How does HUMANr address the subjectivity and potential bias in human judgment?
>
> Please see above.
>
> >Are there plans to adapt the dataset and HUMANr for non-English languages or diverse cultural contexts?
>
> Not at this time, but we would love for others to build on our work. The value of ObjectNet Captions lies in the challenge it poses to models; encouraging models to improve in fundamental ways in order to tackle a more challenging AI problem. We expect the trends in the HUMANr results to hold across languages. As for diverse cultural contexts, the images in ObjectNet were collected from Mechanical Turk workers from across the world from a wide array of cultural and socioeconomic backgrounds. Although not perfect, a cursory look through the dataset will convince you that it represents a broader diversity of contexts than most benchmark datasets.
>
> HUMANr is not language- or dataset-specific! In order to run HUMANr, all one needs is images, reference captions, and candidate captions. The images, references, and candidates can come from any source which makes HUMANr applicable to any and all captioning tasks. We’ve developed and will publicly release a code tool which allows anyone to run HUMANr automatically on any dataset of their choice with a single command line instruction. The code is included in the supplemental material.

---

> ### Author Response · Authors · 2023-11-21
> **Thank you for your review! (Part 4)**
>
> >How can the scalability and cost-effectiveness of HUMANr be improved for widespread adoption?
>
> As discussed in the paper, HUMANr is very cost-effective. A report of HUMANr to benchmark a model in a publication requires, of course, enough images to ensure high confidence which may be over $100 or so, but unofficial runs do not need to be costly. Even just a couple of tens of images for a couple dollars would give a good performance signal for your model. Relative to the cost of building a dataset or training a modern LLM based captioning model these evaluation costs are quite low.
>
> As for usability, we’ve developed a code tool that automatically runs HUMANr evaluation from the command-line. The user only needs to supply the images and captions. The script will post tasks on MTurk and then collect and save HUMANr results. This makes HUMANr not only cheap, but easy to use. The code for this tool is included in the supplemental material.
>
> >Can the authors provide more details on the methodology, especially regarding the computation of HUMANr and the management of subjective human ratings?
>
> Please see response above.

---

### Meta-Review · Area_Chair_Ajre · 2023-12-03

**Metareview:**

This paper introduces ObjectNet Captions, a challenging dataset for image captioning, and presents HUMANr, a new metric for evaluating caption quality. After rebuttal, it received scores of 5556.

The AC agrees with the reviewers that several major concerns still exist. (1) Models like LLaVA and the recent Multimodal LLMs should be tested to make the results convincing. It would be even better to test on GPT4V, on a subset of the images. (2) The dataset focuses on home environments, which limits its generalizability. (3) Human evaluation, especially letting the raters to give a score from 1 to 10 can be very subjective. More discussions are needed to discuss how the authors can make sure the results are reliable and reproducible. Overall, the rebuttal is not convincing enough, and the AC would like to recommend rejection of the paper.

**Justification For Why Not Higher Score:**

The results presented in the paper are not convincing enough, most of the reviewers have shown concerns about the paper, and many of the concerns are not addressed convincingly by the authors.

**Justification For Why Not Lower Score:**

N/A

---

### Decision · Program_Chairs · 2024-01-16

Reject